# An Efficient Synthesis of Acenaphtho[1,2-*b*]indole Derivatives via Domino Reaction

**DOI:** 10.3390/molecules23113045

**Published:** 2018-11-21

**Authors:** Guo-Ning Zhang, Xia Yuan, Weiping Niu, Mei Zhu, Juxian Wang, Yucheng Wang

**Affiliations:** 1Institute of Medicinal Biotechnology, Chinese Academy of Medical Science and Peking Union Medical College, Beijing 100050, China; raisunny2006@163.com (G.-N.Z.); niuweip@126.com (W.N.); mzhu87@163.com (M.Z.); 2State Key Laboratory of Natural and Biomimetic Drugs, School of Pharmaceutical Sciences, Peking University, Beijing 100191, China; yuanxia@bjmu.edu.cn

**Keywords:** acenaphtho[1,2-*b*]indole, domino reaction, enaminone, acenaphthoquinone, l-proline

## Abstract

A concise and efficient synthesis of acenaphtho[1,2-*b*]indole derivatives via the domino reactions of enaminones with acenaphthoquinone catalyzed by l-proline has been developed. This protocol has the advantages of good yields, operational convenience and high regioselectivity.

## 1. Introduction

The indole skeleton is often considered to be one of the most important and fascinating classes of nitrogen-containing heterocycles, and it is often found in both natural products and biologically active compounds [1]. Many indole derivatives have a wide range of biological activities, including anticancer, antioxidant, anti-inflammatory and anti-HIV effects [2,3,4,5,6,7]. In addition, various polycyclic indoles are privileged scaffolds in medicinal chemistry and drug discovery [8,9,10,11]. As a result of these interesting biological activities, many powerful approaches have been developed for the construction of polycyclic indole moieties [12,13,14,15,16,17,18]. However, many of these methods have drawbacks, such as limited availability of starting materials, the use of expensive metal catalysts and the need for harsh reaction conditions. Therefore, developing new and efficient methods for the synthesis of polycyclic indoles and their functionalized derivatives using readily available starting materials is of great importance. Enaminones are commercially available starting materials and have proven to be a useful synthons in the construction of a variety of diverse heterocycles. This synthons has been used in the construction of indole moiety via the condensation with α,β-dicarbonyl compounds under catalyst-free [19,20] or acidic catalyst [21] conditions.

Domino (cascade) reactions are promising and powerful tools in organic and medical chemistry because of their high atom economy, highly complex and diverse products, efficiency in forming multiple bonds, and environmental friendliness [22]. Consequently, domino reactions have often been used for the construction of complex heterocycles [23,24,25,26,27,28]. As part of our program to develop new methods for the construction of important heterocycles by domino reactions [29,30,31,32], we report herein an efficient synthesis of acenaphtho[1,2-*b*]indole derivatives via a domino reaction using l-proline as the catalyst.

## 2. Results and Discussion

We initially evaluated the domino reaction of enaminone **1a** and acenaphthoquinone (**2**). The reaction mixture of **1a** and **2** (1:1 in mole) was subjected to a variety of different conditions and the results are summarized in Table 1. Target product **3a** was obtained in 19% yield when the reaction was carried out under catalyst-free conditions in ethanol at reflux for 2 h followed by dehydroxylation catalyzed by acid (Table 1, entry 1). To our delight, when l-proline (10 mol %) was added, the yield increased to 41% (Table 1, entry 2). Next several other solvents were evaluated for their ability to improve the yield further. The results indicated that toluene was superior to ethanol, chloroform, THF, 1,4-dioxane, DMF, and water in providing much better results (Table 1, entries 2-8). A number of different catalysts were also evaluated for their catalytic efficiency in this reaction. In all cases, the reaction was carried out with 10 mol % of the catalyst in toluene at 80 °C for 2 h. The results revealed that l-proline provided much better results than *p*-TSA, *S*-phenylalanine, phenylalanine, pyrrolidine, piperidine, benzylamine and dibenzylamine (Table 1, entries 9–15). These results indicated that the presence of both secondary nitrogen and a carboxylic acid group plays a crucial role in the desird catalytic activity.

After l-proline had been identified as the best organocatalyst for this reaction, we decided to test the amount of this catalyst required for the full transformation to the desired compounds. The results revealed that when the amount of l-proline increased from 5 mol % to 10 mol %, the yield also increased from 45 to 65% (Table 1, entries 16 and 8). The use of 10 mol % of l-proline in toluene was effective in pushing this reaction forward, and using larger amounts of the catalyst did not improve the yields (Table 1, entries 17–18). The optimization process revealed that the reaction could not proceed in toluene at 40 °C (Table 1, entry 19). To identify the optimum reaction temperature, the reaction was conducted in toluene in the presence of 10 mol % l-proline at 60 °C, 80 °C, and reflux, and these reactions provided product **3a** in yields of 25, 65 and 80% (Table 1, entries 20, 8 and 21), respectively. On the basis of these results, the optimum reaction condition was identified as refluxing with 10 mol % l-proline in toluene for 2 h. Compared with other catalysts (for example, *p*-TSA and TEA), this catalyst has the advantages of higher catalytic efficiency, less toxicity, low cost and ready availability

After the reaction conditions were optimized, the substrate scope of this transformation was also investigated. As shown in Table 2, acenaphthequinone and methyl, bromo, chloro, *t*-Bu and fluoro substituents on the enaminone ring were well tolerated under the reaction conditions, yielding products in satisfactory yields (up to 85%). However, when the enaminones with a bulkier group at the 2- or 2- and 6- positions were used, none of the desired products was obtained (Table 2, entries 15-16).

The structures of compounds **3** were characterized by IR, ^1^H-NMR, and ^13^C-NMR spectra as well as HRMS. The structure of **3g** was further confirmed using single-crystal X-ray diffraction analysis, (Figure 1).

Although details of the mechanism of the domino reaction remain unclear, the formation of compound **3** could be explained by the reaction sequence shown in Scheme 1. The initial reversible reaction of acenaphthoquinone (**2**) with l-proline would give iminium ion **A**. Then, an aza-ene addition of enaminone **1** to iminium ion **A** leads to intermediate **B**, which would undergo a rapid tautomerization to give intermediate **C**. Intermediate **E** would be formed by the intramolecular cyclization of intermediate **C** and the elimination of l-proline. Then, intermediate **F** would be generated by the nucleophilic addition of water to intermediate **E**. In the last step, product **3** would be formed by dehydroxylation of the intermediate catalyzed by H_2_SO_4_ in acetic acid solution.

To support the proposed reaction mechanism, several control experiments were performed (Scheme 2). For example, intermediate **Fa** was obtained in 84% yield from the reaction of **1a** with **2** in refluxing toluene for 2 h catalyzed by 10 mol % l-proline. Desired product **3a** was obtained in 90% yield when intermediate **Fa** was reacted at 80 °C for 2 h in acetic acid catalyzed by H_2_SO_4_.

## 3. Experimental

### 3.1. General Information

All chemicals were obtained commercially and used without further purification. Melting points were measured using an XT-5 micro melting point apparatus from Beijing Tech Instrument Co., Ltd., (Beijing, China) and are uncorrected. NMR spectra were recorded in DMSO-*d*_6_ or CDCl_3_ solution on Inova-300 or 400 MHz spectrometers (Varian, Palo Alto, CA, USA). Chemical shifts values are given in ppm and referred as the internal standard to TMS (tetramethylsilane). The coupling constants (*J*) are reported in hertz (Hz). High-resolution mass spectra (HRMS) were obtained using a MicrOTOF-Q II instrument from Bruker (Billerica, MA, USA). X-ray crystal diffraction analysis was performed with a Mercury CCD X-ray diffractometer (Rigaku, Akishima, Tokyo, Japan).

### 3.2. General Procedure for the Synthesis of Acenaphtho[1,2-b]indole Derivatives 3

A mixture of enaminone **1** (1.0 mmol), acenaphthoquinone (**2**, 1.0 mmol), l-proline (0.1 mmol) and toluene (5 mL) was refluxed for 1–3 h. After the completion of the reaction (confirmed by TLC), the reaction mixture was concentrated in vacuo. Then, acetic acid (15 mL) and conc. H_2_SO_4_ (0.5 mL) were added. The reaction mixture was stirred at 80 °C for 1–2 h. After completion of the reaction (confirmed by TLC), the reaction mixture was then cooled to room temperature and concentrated in vacuo. The crude mixture was purified by column chromatography on silica gel using ethyl acetate/petroleum ether 1:3 as the eluents to give the corresponding product **3**.

*9,9-Dimethyl-7-(p-tolyl)-9,10-dihydro-7H-acenaphtho[1,2-b]indol-11(8H)-one* (**3a**). White solid, *R_f_* = 0.56, m.p. 214–216 °C. IR (KBr, cm^−1^) *ν*: 3063, 2951, 1724, 1630, 1504, 1483, 1323, 1259, 1156, 1140, 860, 771, 722, 679. ^1^H-NMR (400 MHz, CDCl_3_) *δ* 8.11 (d, *J* = 6.4 Hz, 1H, ArH), 7.56 (t, *J* = 8.8 Hz, 2H, ArH), 7.47 (t, *J* = 6.8 Hz, 1H, ArH), 7.35–7.33 (m, 4H, ArH), 7.21 (t, *J* = 7.6 Hz, 1H, ArH), 7.04 (d, *J* = 6.8 Hz, 1H, ArH), 2.56 (s, 2H, CH_2_), 2.42 (s, 3H, CH_3_), 2.40 (s, 2H, CH_2_), 1.05 (s, 6H, 2 × CH_3_). ^13^C-NMR (75 MHz, CDCl_3_) *δ* 192.8, 145.5, 138.1, 137.7, 133.7, 131.0, 130.8, 129.3, 128.2, 128.0, 127.1, 125.6, 125.3, 125.0, 124.7, 124.6, 122.8, 117.9, 115.2, 51.1, 36.1, 34.8, 27.6, 20.3. HRMS (ESI) *m/z*: Calcd. for C_27_H_23_NONa [M + Na]^+^ 400.1677. Found: 400.1705.

*7-(3-Chloro-4-fluorophenyl)-9,9-dimethyl-9,10-dihydro-7H-acenaphtho[1,2-b]indol-11(8H)-one* (**3b**). White solid, *R_f_* = 0.61, m.p. 248–250 °C. IR (KBr, cm^−1^) *ν*: 3027, 2939, 1721, 1494, 1343, 1174, 895, 818. ^1^H-NMR (400 MHz, CDCl_3_) *δ* 8.18 (d, *J* = 6.8 Hz, 1H, ArH), 7.69–7.65 (m, 3H, ArH), 7.56 (t, *J* = 7.6 Hz, 1H, ArH), 7.44–7.31 (m, 3H, ArH), 7.10 (d, *J* = 6.8 Hz, 1H, ArH), 2.57 (s, 2H, CH_2_), 2.43 (s, 2H, CH_2_), 1.12 (s, 6H, 2 × CH_3_). ^13^C-NMR (75 MHz, CDCl_3_) *δ* 193.7, 156.3, 146.2, 138.8, 131.7, 131.6, 129.3, 128.5, 128.2, 128.2, 126.7, 126.5, 126.3, 125.9, 125.8, 124.2, 118.7, 117.9, 117.6, 116.6, 51.9, 37.0, 35.9, 28.6. HRMS (ESI) *m/z*: Calcd. for C_26_H_19_ClFNONa [M + Na]^+^ 438.1037. Found: 438.1020.

*7-(4-Methoxyphenyl)-9,9-dimethyl-9,10-dihydro-7H-acenaphtho[1,2-b]indol-11(8H)-one* (**3c**). White solid, *R_f_* = 0.57, m.p. 240–242 °C. IR (KBr, cm^−1^) *ν*: 3037, 2944, 1723, 1511, 1443, 1078, 816. ^1^H-NMR (400 MHz, CDCl_3_) *δ* 8.10–8.08 (m, 1H, ArH), 7.58–7.54 (m, 2H, ArH), 7.48–7.46 (m, 1H, ArH), 7.37–7.35 (m, 2H, ArH), 7.23–7.17 (m, 1H, ArH), 7.01–7.00 (m, 3H, ArH), 3.84–3.83 (m, 3H, CH_3_O), 2.52 (s, 2H, CH_2_), 2.46 (s, 2H, CH_2_), 1.04 (s, 6H, 2 × CH_3_). ^13^C-NMR (75 MHz, CDCl_3_) *δ* 193.8, 159.6, 146.8, 139.3, 132.1, 131.8, 130.0, 129.2, 129.0, 128.1, 127.1, 126.7, 126.3, 125.9, 125.5, 123.7, 118.8, 116.1, 114.9, 55.6, 52.0, 36.9, 35.7, 28.6. HRMS (ESI) *m/z*: Calcd. for C_27_H_23_NO_2_Na [M + Na]^+^ 416.1626. Found: 416.1629.

*7-(4-Bromophenyl)-9,9-dimethyl-9,10-dihydro-7H-acenaphtho[1,2-b]indol-11(8H)-one* (**3d**). White solid, *R_f_* = 0.60, m.p. 230–232 °C. IR (KBr, cm^−1^) *ν*: 3049, 2952, 1652, 1609, 1494, 1079, 819, 769. ^1^H-NMR (400 MHz, CDCl_3_) *δ* 8.17 (d, *J* = 6.4 Hz, 1H, ArH), 7.73–7.71 (m, 2H, ArH), 7.67–7.62 (m, 2H, ArH), 7.54 (t, *J* = 7.6 Hz, 1H, ArH), 7.41–7.39 (m, 2H, ArH), 7.29 (t, *J* = 7.2 Hz, 1H, ArH), 7.10 (d, *J* = 6.8 Hz, 1H, ArH), 2.55 (s, 2H, CH_2_), 2.41 (s, 2H, CH_2_), 1.10 (s, 6H, 2 × CH_3_). ^13^C-NMR (75 MHz, CDCl_3_) *δ* 193.7, 146.2, 138.6, 136.3, 133.0, 131.8, 131.7, 129.3, 128.7, 128.2, 127.4, 126.7, 126.5, 126.2, 126.1, 124.0, 122.4, 118.9, 116.5, 51.9, 37.0, 35.8, 28.6. HRMS (ESI) *m/z*: Calcd. for C_26_H_20_BrNONa [M + Na]^+^ 464.0626. Found: 464.0633.

*9,9-Dimethyl-7-(4-nitrophenyl)-9,10-dihydro-7H-acenaphtho[1,2-b]indol-11(8H)-one* (**3e**). Yellow solid, *R_f_* = 0.63, m.p. 240–242 °C. IR (KBr, cm^−1^) *ν*: 3036, 2953, 2350, 1728, 1592, 1505, 1329, 1068, 842. ^1^H-NMR (400 MHz, CDCl_3_) *δ* 8.60 (d, *J* = 7.2 Hz, 1H, ArH), 8.52 (d, *J* = 8.4 Hz, 1H, ArH), 8.31–8.26 (m, 3H, ArH), 8.09 (d, *J* = 6.8 Hz, 1H, ArH), 7.84–7.80 (m, 3H, ArH), 7.67 (d, *J* = 8.0 Hz, 1H, ArH), 2.72 (s, 1H, CH_2_), 2.51 (s, 1H, CH_2_), 2.17 (s, 1H, CH_2_), 1.64 (s, 1H, CH_2_), 1.25 (s, 3H, CH_3_), 1.16 (s, 3H, CH_3_). ^13^C-NMR (75 MHz, CDCl_3_) *δ* 187.1, 159.5, 146.1, 145.0, 141.8, 134.3, 132.4, 131.6, 127.5, 127.4, 126.4, 125.8, 125.6, 125.5, 125.3, 124.4, 123.4, 121.0, 117.9, 117.7, 50.9, 35.0, 28.7, 27.6. HRMS (ESI) *m/z*: Calcd. for C_26_H_20_N_2_O_3_Na [M + Na]^+^ 431.1372. Found: 431.1355.

*7-(3,5-Dimethylphenyl)-9,9-dimethyl-9,10-dihydro-7H-acenaphtho[1,2-b]indol-11(8H)-one* (**3f**). White solid, *R_f_* = 0.57, m.p. 220–221 °C. IR (KBr, cm^−1^) *ν*: 3076, 2956, 1728, 1638, 1510, 1474, 1383, 1181, 1080, 870, 836, 801, 767. ^1^H-NMR (400 MHz, CDCl_3_) *δ* 8.18 (d, *J* = 6.4 Hz, 1H, ArH), 7.66–7.62 (m, 2H, ArH), 7.55 (t, *J* = 7.2 Hz, 1H, ArH), 7.30 (t, *J* = 7.6 Hz, 1H, ArH), 7.18–7.15 (m, 3H, ArH), 7.11 (d, *J* = 6.8 Hz, 1H, ArH), 2.65 (s, 2H, CH_2_), 2.48 (s, 2H, CH_2_), 2.46 (s, 6H, 2 × CH_3_), 1.15 (s, 6H, 2 × CH_3_). ^13^C-NMR (75 MHz, CDCl_3_) *δ* 193.9, 146.5, 139.7, 139.1, 137.2, 132.1, 130.3, 129.2, 129.1, 128.1, 126.7, 126.3, 126.0, 123.8, 123.4, 118.9, 116.2, 52.1, 37.2, 35.9, 28.6, 21.4. HRMS (ESI) *m/z*: Calcd. for C_28_H_25_NONa [M + Na]^+^ 414.1834. Found: 414.1847.

*9,9-Dimethyl-7-phenyl-9,10-dihydro-7H-acenaphtho[1,2-b]indol-11(8H)-one* (**3g**). White solid, *R_f_* = 0.55, m.p. 200–202 °C. IR (KBr, cm^−1^) *ν*: 3042, 2958, 1649, 1520, 1500, 1394, 1081, 821, 774, 706. ^1^H-NMR (400 MHz, CDCl_3_) *δ* 8.13 (d, *J* = 6.0 Hz, 1H, ArH), 7.60–7.50 (m, 8H, ArH), 7.23 (dd, *J* = 14.0, 6.8 Hz, 1H, ArH), 7.05 (d, *J* = 6.4 Hz, 1H, ArH), 2.59 (s, 2H, CH_2_), 2.42 (s, 2H, CH_2_), 1.09 (s, 6H, 2 × CH_3_). ^13^C-NMR (75 MHz, CDCl_3_) *δ* 193.8, 146.5, 138.9, 137.3, 132.0, 131.8, 129.8, 129.2, 128.9, 128.7, 128.1, 126.7, 126.3, 126.1, 125.8, 123.8, 118.9, 116.3, 52.0, 37.0, 35.8, 28.6. HRMS (ESI) *m/z*: Calcd. for C_26_H_21_NONa [M + Na]^+^ 386.1521. Found: 386.1503.

*7-(2-Chlorophenyl)-9,9-dimethyl-9,10-dihydro-7H-acenaphtho[1,2-b]indol-11(8H)-one* (**3h**). White solid, *R_f_* = 0.59, m.p. 210–212 °C. IR (KBr, cm^−1^) *ν*: 3041, 2957, 1651, 1518, 1491, 1458, 1069, 821, 771. ^1^H-NMR (400 MHz, CDCl_3_) *δ* 8.15 (d, *J* = 6.8 Hz, 1H, ArH), 7.67–7.59 (m, 3H, ArH), 7.53–7.49 (m, 4H, ArH), 7.23 (d, *J* = 6.4 Hz, 1H, ArH), 6.76 (d, *J* = 6.8 Hz, 1H, ArH), 2.52–2.46 (m, 4H, 2 × CH_2_), 1.14–1.11 (m, 6H, 2 × CH_3_). ^13^C-NMR (75 MHz, CDCl_3_) *δ* 193.8, 147.5, 139.5, 135.1, 132.3, 132.1, 131.7, 131.0, 130.8, 129.3, 129.2, 128.7, 128.2, 128.1, 126.7, 126.4, 126.0, 125.5, 124.0, 118.4, 116.2, 52.2, 36.4, 35.9, 29.0, 28.2. HRMS (ESI) *m/z*: Calcd. for C_26_H_20_ClNONa [M + Na]^+^ 420.1131. Found: 420.1169.

*7-(4-(tert-Butyl)phenyl)-9,9-dimethyl-9,10-dihydro-7H-acenaphtho[1,2-b]indol-11(8H)-one* (**3i**). White solid, *R_f_* = 0.56, m.p. 250–252 °C. IR (KBr, cm^−1^) *ν*: 3021, 2940, 1700, 1452, 1339, 1057, 893, 767. ^1^H-NMR (400 MHz, CDCl_3_) *δ* 8.18 (d, *J* = 5.6 Hz, 1H, ArH), 7.64–7.54 (m, 5H, ArH), 7.47–7.45 (m, 2H, ArH), 7.31–7.29 (t, *J* = 4.8 Hz, 1H, ArH), 7.13 (d, *J* = 6.0 Hz, 1H, ArH), 2.67 (s, 2H CH_2_), 2.48 (s, 2H, CH_2_), 1.44 (s, 9H, C(CH_3_)_3_), 1.14 (s, 6H, 2 × CH_3_). ^13^C-NMR (75 MHz, CDCl_3_) *δ* 193.8, 151.8, 146.6, 139.0, 134.7, 132.0, 131.9, 129.2, 129.1, 128.1, 126.6, 126.3, 126.0, 125.2, 123.8, 119.0, 116.3, 52.1, 37.2, 35.8, 34.9, 31.4, 28.6. HRMS (ESI) *m/z*: Calcd. for C_30_H_28_NO [M − H]^+^ 418.2171. Found: 418.2147.

*7-Phenyl-9,10-dihydro-7H-acenaphtho[1,2-b]indol-11(8H)-one* (**3j**). White solid, *R_f_* = 0.57, m.p. 208–210 °C. IR (KBr, cm^−1^) *ν*: 3038, 2956, 1720, 1698, 1498, 1341, 1136, 837, 734. ^1^H-NMR (400 MHz, CDCl_3_) *δ* 8.10 (d, *J* = 6.0 Hz, 1H, ArH), 7.56–7.43 (m, 8H, ArH), 7.18 (d, *J* = 7.2 Hz, 1H, ArH), 7.01 (d, *J* = 6.0 Hz, 1H, ArH), 2.67 (s, 2H, CH_2_), 2.51 (s, 2H, CH_2_), 2.07 (s, 2H, CH_2_). ^13^C-NMR (75 MHz, CDCl_3_) *δ* 194.4, 147.6, 138.7, 137.4, 132.0, 131.9, 129.8, 129.2, 128.9, 128.6, 128.1, 126.6, 126.4, 126.1, 126.0, 125.7, 123.9, 119.0, 117.4, 38.1, 24.0, 23.3. HRMS(ESI) m/z: Calcd. for C_24_H_16_NO [M − H]^+^ 334.1232. Found 334.1234.

*7-(2-Chlorophenyl)-9,10-dihydro-7H-acenaphtho[1,2-b]indol-11(8H)-one* (**3k**). White solid, *R_f_* = 0.39, m.p. 193–194 °C. IR (KBr, cm^−1^) *ν*: 3051, 2943, 1723, 1656, 1522, 1072, 820, 773, 745. ^1^H-NMR (400 MHz, CDCl_3_) *δ* 8.10 (d, *J* = 6.4 Hz, 1H, ArH), 7.55–7.51 (m, 3H, ArH), 7.47–7.38 (m, 4H, ArH), 7.16 (t, *J* = 7.2 Hz, 1H, ArH), 6.71 (d, *J* = 6.8 Hz, 1H, ArH), 2.61–2.50 (m, 4H, 2 × CH_2_), 2.09–2.05 (m, 2H, CH_2_). ^13^C- NMR (75 MHz, CDCl_3_) *δ* 194.4, 148.6, 139.3, 135.0, 132.1, 131.7, 131.0, 130.8, 129.2, 129.1, 128.7, 128.2, 128.1, 126.7, 126.4, 126.0, 125.7, 123.9, 118.5, 117.4, 38.1, 23.8, 22.6. HRMS (ESI) *m/z*: Calcd. for C_24_H_16_ClNONa [M + Na]^+^ 392.0818. Found: 392.0830.

*7-(2,4-Dimethylphenyl)-9,10-dihydro-7H-acenaphtho[1,2-b]indol-11(8H)-one* (**3l**). White solid, *R_f_* = 0.35, m.p. 264–267 °C. IR (KBr, cm^−1^) *ν*: 3037, 2943, 1724, 1690, 1461, 1337, 1156, 788. ^1^H-NMR (400 MHz, CDCl_3_) *δ* 8.21–8.19 (m, 1H, ArH), 7.66–7.56 (m, 5H, ArH), 7.51–7.44 (m, 2H, ArH), 7.31 (t, 1H, *J* = 4.8 Hz, ArH), 7.15 (d, *J* = 6.0 Hz, 1H, ArH), 2.84–2.79 (m, 2H, CH_2_), 2.64–2.58 (m, 2H, CH_2_), 2.22–2.15 (m, 2H, CH_2_), 1.42 (s, 9H, 3 × CH_3_). ^13^C-NMR (75 MHz, CDCl_3_) *δ* 194.5, 151.8, 147.8, 138.7, 134.6, 132.1, 131.9, 129.2, 129.0, 128.1, 126.6, 126.3, 126.0, 125.1, 123.8, 119.1, 117.3, 38.1, 34.9, 31.4, 24.0, 23.4. HRMS calcd for C_28_H_25_NONa [M + Na]^+^ 414.1834, found 414.1834.

*7-(4-(tert-Butyl)phenyl)-9,10-dihydro-7H-acenaphtho[1,2-b]indol-11(8H)-one* (**3m**). White solid, *R_f_* = 0.36, m.p. 190–192 °C. IR (KBr, cm^−1^) *ν*: 3049, 2953, 1655, 1521, 1082, 819, 773. ^1^H-NMR (400 MHz, CDCl_3_) *δ* 8.17 (d, *J* = 6.4 Hz, 1H, ArH), 7.64–7.59 (m, 2H, ArH), 7.53 (t, *J* = 7.2 Hz, 1H, ArH), 7.25–7.22 (m, 3H, ArH), 7.17 (d, *J* = 8.0 Hz, 1H, ArH), 6.76 (d, *J* = 6.8 Hz, 1H, ArH), 2.67 (t, *J* = 10.4 Hz, 1H, CH_2_), 2.62–2.59 (m, 2H, CH_2_), 2.53 (t, *J* = 5.6 Hz, 1H, CH_2_), 2.44 (s, 3H, CH_3_), 2.18–2.15 (m, 2H, CH_2_), 2.07 (s, 3H, CH_3_). ^13^C-NMR (75 MHz, CDCl_3_) *δ* 194.3, 148.1, 139.5, 135.2, 133.7, 132.3, 132.1, 131.8, 129.2, 129.0, 128.2, 127.9, 127.2, 126.7, 126.2, 125.8, 123.7, 118.3, 117.0, 38.2, 24.0, 22.6, 21.3, 17.4. HRMS (ESI) *m/z*: Calcd. for C_26_H_21_NONa [M + Na]^+^ 386.1521. Found: 386.1566.

*7-(3,5-Dimethylphenyl)-9,10-dihydro-7H-acenaphtho[1,2-b]indol-11(8H)-one* (**3n**). White solid, *R_f_* = 0.34, m.p. 222–224. IR (KBr, cm^−1^) *ν*: 3043, 2940, 1721, 1660, 1521, 1071, 1034, 852, 815, 767. ^1^H-NMR (400 MHz, CDCl_3_) *δ* 8.19–8.17 (m, 1H, ArH), 7.66–7.62 (m, 2H, ArH), 7.56–7.51 (m, 1H, ArH), 7.31–7.28 (m, 1H, ArH), 7.17–7.11 (m, 4H, ArH), 2.81 (t, *J* = 5.2 Hz, 2H, CH_2_), 2.63–2.59 (m, 2H, CH_2_), 2.44 (s, 6H, 2 × CH_3_), 2.19 (t, *J* = 5.6 Hz, 2H, CH_2_). ^13^C-NMR (75 MHz, CDCl_3_) *δ* 194.5, 147.7, 139.6, 138.8, 137.2, 132.1, 131.9, 130.2, 129.2, 129.1, 128.1, 126.7, 126.3, 126.0, 123.8, 123.3, 118.9, 117.3, 38.1, 24.0, 23.4, 21.4. HRMS (ESI) *m/z*: Calcd. for C_26_H_20_NO [M − H]^+^ 363.1545. Found: 363.1557.

### 3.3. General Procedure for the Synthesis of Tetrahydroacenaphtho[1,2-b]indole Derivatives **Fa**

A mixture of enaminone (**1a**) (1.0 mmol), acenaphthoquinone (**2**) (1.0 mmol), l-proline (0.1 mmol) and toluene (5 mL) was refluxed for 3 h. After the completion of the reaction (confirmed by TLC), the reaction mixture was concentrated in vacuo. The crude mixture was purified by column chromatography on silica gel using ethyl acetate/petroleum ether 1:1 as the eluents to give corresponding product **Fa**. 

*6b,11b-Dihydroxy-9,9-dimethyl-7-(p-tolyl)-8,9,10,11b-tetrahydro-6bH-acenaphtho[1,2-b]indol-11(7H)-one* (**Fa**). White solid, *R_f_* = 0.23, m.p. 240–242 °C. IR (KBr, cm^−1^) *ν*: 3598, 2961, 2870, 1790, 1606, 1511, 1436, 1406, 1283, 1038, 783. ^1^H NMR (300 MHz, DMSO-*d*_6_) *δ* 7.94–6.93 (m, 10H, ArH), 6.46 (s, 1H, OH), 5.80 (s, 1H, OH), 2.38–1.79 (m, 7H, CH_3_ + 2 × CH_2_), 1.01 (s, 3H, CH_3_), 0.83 (s, 3H, CH_3_). ^13^C NMR (100 MHz, DMSO-*d*_6_) *δ* 194.5, 168.6, 149.7, 145.7, 142.4, 140.9, 139.8, 136.0, 135.0, 134.9, 134.2, 133.8, 132.7, 130.1, 128.6, 126. 6, 124.8, 115.9, 107.8, 91.9, 56.3, 42.3, 38.7, 34.7, 32.3, 26.2. HRMS (ESI) *m*/*z*: Calcd. for C_27_H_25_NO_3_Na [M + Na]^+^ 434.1732. Found: 434.1734.

## 4. Conclusions

In summary, we have developed an efficient protocol for the construction of acenaphtho[1,2-*b*]indole derivatives via the domino reaction of enaminones with acenaphthoquinone catalyzed by l-proline. This protocol has the advantages of mild reaction conditions, high yields and operational convenience.

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
