# Peer review of "An Efficient Synthesis of Acenaphtho[1,2-b]indole Derivatives via Domino Reaction"

_molecules, 2018, doi:10.3390/molecules23113045_

Reviewer 1 Report

The work is a continuation of previous studies [notes 37-40]. Though the reactions of enaminones with α,β-dicarbonyl compounds  are well known the authors have elaborated their original procedure using L-proline as a catalyst and synthesized 14 new compounds.  The manuscript could be published only after  major revision of the whole text.

1) The introduction is too general and useless in terms of  structure-activity relationship information. The first  statement about nitrogen-containing heterocycles is obvious and superfluous.  I suggest to change introduction by its shortening and including information only about relatively close condensed indoles (e.g. Indeno[1,2-b]indole-9,10-diones)  providing literature data about reaction condition variations for condensation of enaminones with α,β-dicarbonyl compounds.

2) Solvents used for chromatographic purification of each product with Rf values should be given.

3) Preparative procedure and physico-chemical properties of intermediate compound Fa are not described.  Please, provide full experimental information for scheme 2.

 4) Table 2: there is no difference between compounds 3j and 3m, compound 3m in the experimental part  correspond with compound 3l in table 3. The Structure of Compound 3l  does not confirm its name. I advise to go over all experimental part and all structures and correct any mistakes.

Author Response

1. The introduction is too general and useless in terms of structure-activity relationship information. The first statement about nitrogen-containing heterocycles is obvious and superfluous. I suggest to change introduction by its shortening and including information only about relative close condensed indoles (e.g. indeno[1,2-b]indole-9,10-diones) providing literature data about reaction conduction variations for condensation of enaminones with a,β-dicarbonyl compounds.

Reply: This is a good suggestion. The introduction has been shortened in the revised manuscript.

2. Solvent used for chromatographic purification of each product with Rf values should be given.

Reply: According to the reviewer’s suggestion, the solvents and Rf values have been added in the revised manuscript.

3. Preparative procedure and physic-chemical properties of intermediate compound Fa are not described. Please provide full experimental information for Scheme 2.

Reply: According to the reviewer’s suggestion, the synthetic procedure and physic-chemical properties of intermediate compound Fa have been added in the revised manuscript.

4. Table 2, there is no difference between compounds 3j and 3m, compound 3m in the experimental part correspond with compound 3l in table 3. The structure of compound 3l does not confirm its name. I advise to go over all experimental part and all structures and correct any mistakes.

Reply: I am very sorry for there is some typographical error in the manuscript. These mistakes have been corrected in the revised manuscript.

Reviewer 2 Report

The article in question describes a L-proline catalyzed cascade reaction between enaminones and acenaphtoquinone. The authors do a nice job of highlighting the importance of indole derivatives and other N-heterocycles, but don’t specify any particular utility of the target compounds in this study. The structures are definitely interesting, but do they have any real value? While Table 1 is well presented and discussed, I would like to have seen the authors investigate other amine sources as catalysts. In other words, I am not convinced by this table that there is anything specific to L-proline that it makes it the ideal amine catalyst. Only one alternative amine is tested, and aniline is not a great choice due to the reduced nucleophilicity of the nitrogen atom. I am also confused by entry 10. This is probably just a nomenclature issue, but how is S-Proline different than L-Proline? The chiral center is L-proline is an S center. However, even if it was the enantiomer of L-proline, the stereochemistry of the chiral center should have no impact on these achiral reactants, so it is unclear to me why the yield is so limited in this trial. It would useful to know how pyrrolidine, piperidine, benzylamine, dibenzylamine, or others would behave in these reactions while adjusting the pH. If the L-proline is only serving to generate the activated iminium electrophile as is proposed in the article, then these other amines should be equally useful in the reaction. They are also more readily soluble in toluene, and are all significantly cheaper than L-proline. Table 2 shows that the reaction tolerates both electron-rich and electron-poor aromatic rings and provides reasonable to good yields for all of the presented substrates. It would be nice to include a few examples including with some bulkier groups at the 2 or 2 and 6 positions in order to give some idea of when sterics begin to limit the cyclization. The article is generally well-written and easy to follow. In summary, the authors have demonstrated a novel route to some interesting structures. My major complaint is that a few control reactions are missing to make it clear that the reaction conditions have been fully optimized.

 Author Response

1. The authors do a nice job of highlighting the importance if indole derivativws and other N-hetercycles, but don’t specify any particular utility of the target compounds in this study. The structures are definitely interesting, but do they have any real value?

Reply: The indole skeleton has been reported having a wide range of biological activities in the literature. But the biological activities of this title compounds is not clear.

2. I would like to have seen the authors investigate other amine sources as catalyst in other words. I am not convinced by this table that is anything specific to L-proline that it makes it the ideal amine catalyst. Only alternative amine is tested and aniline is not a great choice due to the reduced nucleophilicity of the nitrogen atom. I am also confused by entry 10. This is probably just a nomenclature issue, but how is S-proline different than L-proline? The chiral center is L-proline is an S center. However, even if it was the enantiomer of L-proline, the stereochemistry of the chiral center should have no impact on these achiral reactions, so it is unclear to me why the yield is so limited in this trial. It would useful to know how pyrrolidine, piperidine, benzylzmine, dibenzylamine, or others would behave in these reactions while adjusting in the pH. If the L-proline is only serving  to generate the activated iminium electrophile as is proposed in the article, then these other amines should be equally useful in the reaction. They are also more readily soluble in toluene, and are all significantly cheaper than L-proline.

Reply: I am very sorry for there are some typographical errors in the manuscript. In table 1, entry 10, the “S-proline” should be “S-phenylalanine” and it has been corrected in the revised manuscript. According to the reviewer’s suggestion, some amines such as pyrrolidine, piperidine, benzylzmine, dibenzylamine have been used as catalysts and the results have been added in Table 1 entries 12-15.

3. Table 2 shows that the reaction tolerates both electron-rich and electron-poor aromatic rings and provides reasonable to good yields for all of the presented substrates. It would be nice to include a few examples including with some bulkier groups at the 2 or 2 and 6 positions in order to give some idea of when sterics begin to limit the cyclization.

Reply: According to the reviewer’s suggestion, two enaminones with some bulkier groups at the 2 or 2 and 6 positions have been used in this reaction. However, there are no desired products were obtained. The results have been added in Table 2 in this revised manuscript.

Reviewer 3 Report

The present manuscript (Manuscript ID: PC-15-1145) refers to new and interesting results concerning the An Efficient Synthesis of Acenaphtho[1,2-b]indole Derivatives via Domino Reaction. This work is an efficient catalyst in many organic reactions. In addition, This results are very well explained and the paper technically correct. Lastly, this paper can be a very good contribution to Molecules.

Author Response

The present manuscript (Manuscript ID: PC-15-1145) refers to new and interesting results concerning the An Efficient Synthesis of Acenaphtho[1,2-b]indole Derivatives via Domino Reaction. This work is an efficient catalyst in many organic reactions. In addition, These results are very well explained and the paper technically correct. Lastly, this paper can be a very good contribution to Molecules.

Reply:  

We thank the reviewer for the positive and encouraging evaluation of our work.

Reviewer 4 Report

table 1 footnote change for " All the reactions"

L85-86 carry out is not correct

Author Response

1. Table 1 footnote change for “All the reactions”.

Reply: In the revised manuscript, the footnote for table 1 has been added.

2. L85-86 carry out is not correct.

Reply: According to the reviewer’s suggestion, the “carry out” has been changed in the line 85-86.

Round  2

Reviewer 1 Report

The authors corrected almost all notes, therefore I recommend the manuscript for publication. Nevertherless I suggest to add information about other synthetic procedures for enaminones with a,β-dicarbonyl compounds yielding indole moiety, e.g. noncatalysed reaction (Journal of Chemical Research, Synopses, (8), 244-5; 1985; Synlett, (20), 3484-3488; 2006) or acidic catalysis (Tetrahedron, 70(31), 4595-4601; 2014). The procedure with triethylamine or pTSA as catalysts published in your previous work [16] seems practical and easy-to-use (yields were 82-93%), so what was the reason to seach for another more expensive type of the catalyst? Please, provide any advantages of using L-proline.

Author Response

To Reviewer’s comments:

1. The authors corrected almost all notes, therefore I recommend the manuscript for publication. Nevertherless I suggest to add information about other synthetic procedures for enaminones with a,β-dicarbonyl compounds yielding indole moiety, e.g. noncatalysed reaction (Journal of Chemical Research, Synopses, (8), 244-5; 1985; Synlett, (20), 3484-3488; 2006) or acidic catalysis (Tetrahedron, 70(31), 4595-4601; 2014).

Reply: According to the reviewer’s suggestion, The information about other synthetic methods using enaminones with a,β-dicarbonyl compounds yielding indole moiety have been added in the revised manuscript (page 1, lines 30-33 and references 19-21).

2. The procedure with triethylamine or p-TSA as catalysts published in your previous work [16] seems practical and easy-to-use (yields were 82-93%), so what was the reason to search for another more expensive type of the catalyst? Please, provide any advantages of using L-proline.

Reply: This is a good suggestion. In the literature, p-TSA and TEA can give a good yield. In our paper, the desired products were obtained by two steps (catalyzed by L-proline and catalyzed H2SO4, respectively), using the L-proline can give>90% yield. According to the reviewer’s suggestion, some advantages of using L-proline have been given in the revised manuscript (Page 2, lines 66-68). In addition, the paper [reference 16] is not our previous work.